# Biomarker-Based Nomogram to Predict Neoadjuvant Chemotherapy Response in Muscle-Invasive Bladder Cancer

**DOI:** 10.3390/biomedicines13030740

**Published:** 2025-03-18

**Authors:** Meritxell Pérez, Juan José Lozano, Mercedes Ingelmo-Torres, Montserrat Domenech, Caterina Fernández Ramón, J. Alfred Witjes, Antoine G. van der Heijden, Maria José Requena, Antonio Coy, Ricard Calderon, Begoña Mellado, Antonio Alcaraz, Antoni Vilaseca, Maria J. Ribal

**Affiliations:** 1Department of Urology, Hospital Universitari Terrassa, 08221 Terrassa, Spain; mperez@cst.cat; 2Plataforma Bioinformatica, Centro de Investigación Biomédica en Red Enfermedades Hepáticas y Digestivas (CIBERehd), Hospital Clínic de Barcelona, 08036 Barcelona, Spain; 3Department of Urology, Institut d’Investigacions Biomèdiques August Pi i Sunyer (IDIBAPS), Hospital Clínic de Barcelona, University of Barcelona, 08036 Barcelona, Spain; 4Medical Oncology Department, Fundació Althaia, Xarxa Assintencial Universitària de Manresa, 08242 Manresa, Spain; 5Urology Department, Fundació Althaia, Xarxa Assintencial Universitària de Manresa, 08242 Manresa, Spain; 6Department of Urology, Radboud University Medical Center, 6525 GA Nijmegen, The Netherlands; 7Department of Urology, Reina Sofía University Hospital, IMIBIC, Cordoba University, 14014 Córdoba, Spain; 8Fundación Instituto Valenciano de Oncologia (IVO), 46009 Valencia, Spain; 9Translational Genomics and Targeted Therapeutics in Solid Tumors, Medical Oncology Department, August Pi i Sunyer Biomedical Research Institute (IDIBAPS), Hospital Clínic de Barcelona, 08036 Barcelona, Spain; 10Uro-Oncology Unit, Medical Oncology Department, Hospital Clínic de Barcelona, University of Barcelona, 08036 Barcelona, Spain; 11Uro-Oncology Unit, Department of Urology, Hospital Clinic de Barcelona, 08036 Barcelona, Spain

**Keywords:** bladder cancer, gene expression profile, neoadjuvant chemotherapy, prognostic model, response prediction

## Abstract

**Background/Objectives:** The aim of this study was to identify response prediction and prognostic biomarkers in muscle-invasive bladder cancer (MIBC) patients undergoing neoadjuvant chemotherapy (NAC). **Methods**: A retrospective multicentre study including 191 patients with MIBC who received NAC previous to radical cystectomy (RC) between 1996 and 2013. Gene expression patterns were analysed in 34 samples from transurethral resection of the bladder (TURB) using Illumina microarrays. The expression levels of 45 selected differentially expressed genes between responders and non-responders to NAC were validated by quantitative PCR in an independent cohort of 157 patients. Regression analysis was used to identify predictors of downstaging and relapse. A nomogram for predicting downstaging and relapse—including clinicopathological and gene expression variables—was developed. **Results**: The expression levels of 1352 transcripts differed between responders and non-responders to NAC. A nomogram based on the most predictive clinical variables (age, Tis (in situ), gender, history of NMIBC, and lymphadenopathy) and genes selected following the Akaike information criterion (AIC) (CBTB16, CHMP6, DDX54, CASP8, LOR, and PLEC) was then created. In addition, a three-gene expression prognostic model to predict tumour relapse was generated. This model was able to discriminate between two groups of patients with a significantly different probability of tumour relapse (HR: 2.11; CI: 1.16–3.83, *p* = 0.01). **Conclusions**: Our nomogram based on gene expression and clinical data is a useful tool to predict downstaging and tumour relapse after NAC in MIBC patients. Further validation is warranted.

## 1. Introduction

Bladder cancer represents the most frequent malignant condition of the urinary tract. It has an annual incidence of approximately 550,000 new cases worldwide, with around 200,000 deaths each year. The incidence among males is markedly higher than that of females, with males accounting for approximately 75% of newly diagnosed cases [1]. Approximately 25% of bladder cancers have detrusor muscle invasion, which has a high progression rate to metastasis. In fact, half of these patients may die within 2 years from diagnosis despite undergoing radical cystectomy (RC) [2].

Neoadjuvant cisplatin-containing combination chemotherapy is strongly recommended for treating T2-T4N0M0 MIBC, which are tumours invading the muscle layer, the fatty tissue above the bladder muscle, or the regional organs outside the bladder (prostate, uterus, vagina, and wall of the pelvis or of the abdomen). Three meta-analyses were conducted to assess whether neoadjuvant chemotherapy (NAC) improves life expectancy. One comprehensive meta-analysis, incorporating revised data from 11 randomised studies involving 3005 patients, demonstrated a notable survival advantage with NAC. A subsequent meta-analysis included four additional randomised controlled trials (RCTs), incorporating fresh data from 427 patients and updated results from 1596 participants in the Nordic I, Nordic II, and BA06 30,894 trials. This analysis confirmed earlier findings, showing an 8% absolute increase in five-year survival rates, a need-to-treat ratio of 12.5, and a 30–40% complete pathological response [3]. Compared to the traditional regimens of either cisplatin + gemcitabine (CG) or [cisplatin + gemcitabine or methotrexate + vinblastine + doxorubicin + cisplatin (MVAC)] for four cycles, new chemotherapy regimens have shown a little benefit in terms of complete response rates at the expense of increased toxicity. The VESPER trial compared four cycles of CG versus six cycles of dose-dense MVAC (ddMVAC). The complete pathological response improved from 36% to 42% [4].

However, MIBC patients who do not respond to NAC may experience delays in surgical treatment and face potential chemotherapy side effects. In some series, a delay of more than 12 weeks between diagnosing MIBC and performing RC is clearly associated with a worse prognosis for patients not treated with NAC [5].

Currently, there are no demographic, clinical, or biomarker predictors of the response to NAC that can be recommended for treatment decisions and defining the patient population that will benefit from systemic treatment [6]. Recently, some molecular classifications have been proposed to identify MIBC patients who are likely to respond to chemotherapy but the diversity of subtype sets hinders their clinical application. Notably, the Bladder Cancer Molecular Taxonomy Group has proposed six MIBC molecular subtypes, which differ in underlying oncogenic mechanisms, immune and stromal cell infiltration, and histological and clinical characteristics. For instance, they suggest that one subtype (basal) may respond better to chemotherapy. Although promising, this still requires validation [7]. Therefore, developing a tool to predict the response of MIBC patients to NAC and their subsequent relapse would be highly beneficial in the clinical setting.

Gene expression patterns enable comprehensive analyses of complex molecular activities in cancer cells, potentially identifying responders and non-responders to NAC before treatment. Some authors have previously described the potential usefulness of gene expression profiles in predicting response to NAC in various solid tumours [8,9,10,11,12], including bladder cancer [13,14]. Other predictive biomarkers, such as DNA-repair-associated genes ERC22, MRE11, ATM, RB1, and FANCC, have also been investigated for their potential to predict the response to cisplatin-based chemotherapy [15,16].

In this study, we performed a comprehensive analysis of gene expression patterns in tumours from MIBC patients, both responders and non-responders to NAC, to identify the molecular biomarkers of downstaging and tumour relapse. Our goal was to develop a tool to predict which MIBC patients will respond to NAC and the subsequent relapse. Additionally, we aimed to integrate molecular and clinical data to develop a feasible and useful nomogram that accurately predicts the response to NAC in MIBC. By achieving this, we hope to significantly enhance the management and treatment outcomes of MIBC.

## 2. Materials and Methods

### 2.1. Patients and Samples

We conducted a multicentre retrospective study including 191 patients with MIBC who received NAC previous to RC between 1996 and 2013 at four centres (Hospital Clínic Barcelona, Barcelona, Spain; Erasmus University Medical Center, Rotterdam, The Netherlands; Hospital Sant Joan de Déu, Manresa, Spain; Hospital Reina Sofia, Cordoba, Spain). Inclusion criteria comprised all patients undergoing RC after NAC with clinical and pathological information available, and with sufficient tissue to perform the genetic testing. After tissue testing, 42 samples were excluded from the final analysis, leaving a final sample size of 149 (see sample exclusion criteria below in Section 2.4 on the validation phase).

The NAC schedule consists of the intravenous administration of CG or MVAC for four cycles. All patients were followed up postoperatively according to the current EAU guidelines [17]. Briefly, CT scans were performed at 3 months after surgery, every 6 months for the first 3 years, annually until 5 years, and biannually thereafter. Response was defined as a downstaging in the T stage to <T1 in the cystectomy piece relative to the TURB sample after NAC. Relapse was defined as distant metastasis, pathological nodes, or local recurrence during follow-up. Follow-up time started on the date of surgery, and observations were censored on the date of the last follow-up. The Hospital Clinic of Barcelona’s Institutional Review Board approved this study (HCB2007/3560).

### 2.2. Tissue Specimens and RNA Isolation

Once obtained, tissue was fixed in 10% formalin within 24 h and subsequently embedded in paraffin. A slide of each specimen was stained with haematoxylin and eosin to determine the presence of tumour cells. Formalin-fixed paraffin-embedded (FFPE) tissue blocks were reviewed by a single pathologist. RNA was isolated from FFPE macrodissected specimens (total thickness: 80 mm) using the RecoverAll Total Nucleic Acid Isolation Kit for FFPE (Ambion Inc., Austin, TX, USA) according to the manufacturer’s protocol. Total RNA was quantified by spectrophotometric analysis at 260 nm (NanoDrop Technologies, Wilmington, DE, USA).

### 2.3. Whole-Genome Gene Expression Microarray: Biomarker Discovery Phase

A flowchart of the entire study is shown in Figure 1. In order to identify specific differentially expressed genes in responders vs. non-responders to NAC, global expression patterns were analysed in 34 TURB samples (17 responders and 17 non-responders) using a Whole-Genome Gene Expression DASL HT assay (Illumina, San Diego, CA, USA) according to manufacturer’s instructions.

### 2.4. Reverse-Transcription Quantitative PCR (RT-qPCR): Biomarker Validation Phase

Differential expression of 45 genes selected from the microarray analysis were selected to be validated in an additional set of 157 tissue TURB samples from MIBC patients who underwent NAC using BioMark 48.48 Dynamic arrays (Fluidigm, South San Francisco, CA, USA). Genes were considered differentially expressed when the False Discovery Rate (FDR) was ≤0.1 and absolute fold change (FC) value ≥ 1.5. TaqMan Gene Expression assays 20X (Thermofisher, Waltham, MA, USA) from the 45 target genes and the two endogenous controls were loaded into the BioMark 48.48 Dynamic array following the manufacturer’s instructions. Real-time qPCR analysis software was used to obtain the cycle quantification (Cq) values. The threshold was automatically calculated for each gene and the relative expression levels of target genes within a sample were expressed as DCq (DCq = Cq_endogenous control_ − Cq _target gene_). The geometric mean Cq value of PPIA and BGUS was used as an endogenous control. Genes (n = 19) expressed in less than 50 samples were excluded. Furthermore, samples (n = 21) with less than 35 genes with valid expression values and samples (n = 21) not expressing BGUS were excluded. Thus, 115 samples and 26 genes remained valid for further analyses.

### 2.5. Downstaging Predictors

Univariate and multivariate forward stepwise binary logistic regression analysis was used to evaluate the ability of the 26 differentially expressed target genes to predict treatment response. We made a nomogram based on all the variables by using a step function on R that minimises the number of variables of the model while maintaining their agreement as much as possible.

This model uses 10 variables in total, the clinical information of age, sex, adenopathy, and history of NMIBC and, in addition, the information of a six-gene signature.

### 2.6. Survival Analysis

Univariate and multivariate forward stepwise Cox’s regression analysis was used on the validated differentially expressed target genes to investigate their ability to predict tumour progression. After establishing the multivariate model for relapse, a risk score for the variables of the model was calculated for each patient. The risk score was subjected to ROC analysis to choose the most appropriate threshold for predicting relapse; thereafter, Kaplan–Meier curves were generated using the selected thresholds and compared according to the log-rank test.

Statistical significance was defined at a *p* value of 0.05.

To create the nomogram, the predictive clinical variables were used together with a selection of 6 genes’ expression levels. The differentially expressed genes were selected using the Akaike information criterion (AIC), an estimator of prediction error that estimates the relative amount of information lost by a given model and selects the most accurate model using the minimum number of variables included.

Data were processed in the R statistical environment (v3.3.2).

## 3. Results

### 3.1. Clinicopathological Features of the Cohort

The clinicopathological characteristics of the MIBC patients, split by study phases, are listed in Table 1.

A total of 44% (51/115) patients from the biomarker validation phase presented downstaging after NAC. During the follow-up period (median follow-up: 1.88; range: 0.1–15.87 years), a total of 50 patients (53.7%) developed tumour progression. The median time to disease progression was 0.66 years (range: 0.1–4.85).

### 3.2. Biomarker Discovery Phase

Overall, the analysis of the gene expression profile of patients with MIBC who underwent NAC prior to cystectomy resulted in the identification of 1352 transcripts differentially expressed between responders and non-responders (FDR < 0.1), with 749 upregulated and 603 downregulated transcripts in responding patients.

### 3.3. Classifier Development Phase

The expression levels of 26 selected genes, following the same method as described previously by our group for the microarray analysis, were successfully determined in an independent cohort of 115 MIBC patients who underwent NAC [18].

Cox’s regression analysis was used to evaluate each gene as a downstaging identifier. Then, a stepwise regression in R was performed, which consists of iteratively adding and removing predictors to find the subset of variables in the dataset, resulting in the best predictive model maintaining the concordance using fewer variables.

Clinical variables (age, gender, history of NMIBC, and clinical lymph node involvement) and a six-expression classifier following the Akaike information criterion (AIC) (CBTB16, CHMP6, DDX54, KASP8, LOR, and PLEC) (Table 2) were used to define our downstaging risk nomogram with a concordance of 0.75 (SD: 0.04) (Figure 2).

Furthermore, Cox’s regression analysis showed that FN1, VASP, and CEP63 were independent prognostic factors of tumour relapse (Table 3). Additionally, a risk score (RS) for predicting tumour relapse after NAC and cystectomy was calculated for each patient according to a mathematical algorithm containing FN1, VASP, and CEP63. The median range value of this risk score was 0.85 (0.73–1). Thereafter, an ROC analysis of this combined gene expression model allowed for the selection of a threshold of 1.211 to classify patients into a high- and low-risk group for relapse. The combined model in classifying patients into a high- and low-risk group for relapsing after NAC treatment shows an AUC of 0.629 (Figure 3), suggesting a limited capability of discrimination.

Figure 4 shows the Kaplan–Meier curve of the combined classifier generated using the selected threshold. As shown, a RS generated using gene expression values was able to discriminate between the two groups of patients with a significantly different probability of relapse (hazard ratio (HR): 2.11; confidence interval (CI): 1.16–3.83, *p* = 0.01).

## 4. Discussion

RC with pelvic lymphadenectomy remains the gold standard for treating MIBC. However, variability in the clinical outcomes among patients with similar pathological stages and lymph node statuses highlights the limitations of current prognostic models and underscores the need for more refined predictive tools that capture biological heterogeneity [19]. Various factors (T3b-T4 disease, hydroureteronephrosis and/or histological evidence of lymphovascular invasion, and neuroendocrine or micropapillary features) have been associated with the chemosensitivity of bladder cancer. However, these factors have failed to predict individual responses with high accuracy [20].

Recent advancements in molecular biology have led to the identification of various biomarkers that enhance diagnostic, prognostic, and treatment prediction strategies. Our study addresses a critical gap by developing a nomogram that integrates both genetic and clinical predictors to forecast responses to NAC. This tool aims to refine treatment strategies, enhancing the precision of therapeutic decisions and potentially reducing the incidence of adverse effects and unnecessary delays in surgery for non-responders.

In recent years, tremendous advances have been made in the discovery of new markers associated with molecular alterations, showing considerable clinical relevance in diagnosis, tumour classification, prognosis, and prediction of an individual patient’s response to treatment. However, validation of these genetic and molecular markers could improve the prediction of an individual’s response to NAC, providing useful information for characterising the nature of individual cancers and identifying clues to distinguish those tumours showing a good response to certain chemotherapy from those showing a poor response [15,16].

The establishment of an accurate prognostic classifier has important clinical implications. For instance, if we know in advance that a patient will not respond to NAC, we could directly offer the only potentially curative treatment in these patients and go directly to RC.

This study was designed to identify biomarkers to determine MIBC patients who will respond to NAC, both in terms of downstaging and survival benefit. We first identified and validated differentially expressed genes of TURB samples from responders and non-responders to NAC. Subsequently, we explored the influence of identified genes in downstaging and tumour relapse. In addition, we included clinical variables in our nomogram to increase its capability of identifying responders.

By incorporating genes like KASP8 and PLEC, which are crucial in other cancer cell dynamics and chemotherapy responses, the nomogram provides a robust prediction model for treatment outcomes [21,22,23,24,25,26,27,28,29,30,31,32,33,34,35]. In addition, CBTB16 upregulation by activation of the glucocorticoid receptor plays a crucial role in cancer biology, encoding for a transcriptional suppressor that controls growth in ER-positive breast cancer and correlates with prognosis in luminal A patients [36]. Also, CBTB16 has been associated as a biomarker for primitive neuroectodermal tumour element/Ewing sarcoma [37] and for yolk sac tumours [38].

Including information from the proposed nomogram allows clinicians to make more informed decisions regarding patient care. The nomogram offers individualised risk assessments, enabling tailored treatment plans. This personalisation helps avoid over-treatment and minimises unnecessary side effects, thereby improving patient quality of life.

Our study also introduces a RS that integrates patient demographics, tumour characteristics, and key gene expressions such as FN1, VASP, and CEP63 to predict tumour relapse in MIBC patients undergoing NAC. This tool’s clinical relevance is highlighted by its ability to enhance predictive accuracy, personalise treatment plans, validate biological markers, optimise therapeutic efficacy, and minimise treatment-related burdens.

While our nomogram shows promise, its clinical application requires further validation in prospective cohorts. This step is crucial to confirm predictive accuracy, ensuring the nomogram’s predictions are reliable and applicable in diverse patient populations. Additionally, addressing limitations such as the retrospective study design and small sample size will enhance the model’s robustness. Demonstrating the nomogram’s effectiveness in real-world clinical settings is essential to gain broader acceptance and implementation.

In conclusion, our nomogram represents a significant advancement in the personalised management of bladder cancer. By providing precise, individualised risk assessments, it aims to optimise therapeutic efficacy and minimise treatment-related burdens, ultimately improving patient care and outcomes. Further validation will solidify its role in clinical practice, making it an invaluable tool for the medical community.

## 5. Conclusions

Our prognostic model based on differential gene expression and clinical variables is able to discriminate between responders and non-responders with MIBC before undergoing NAC.

Our study is a step forward in the identification of predictive and prognostic models of MIBC patients undergoing NAC that can potentially be implemented in daily practice in the future if validated in further studies. Its future applicability would be an important step forward for the individualisation of treatment in bladder cancer. Eventually, cost-effectiveness analyses will be necessary to define the role of this nomogram in clinical practice.

## Figures and Tables

**Figure 1 biomedicines-13-00740-f001:**
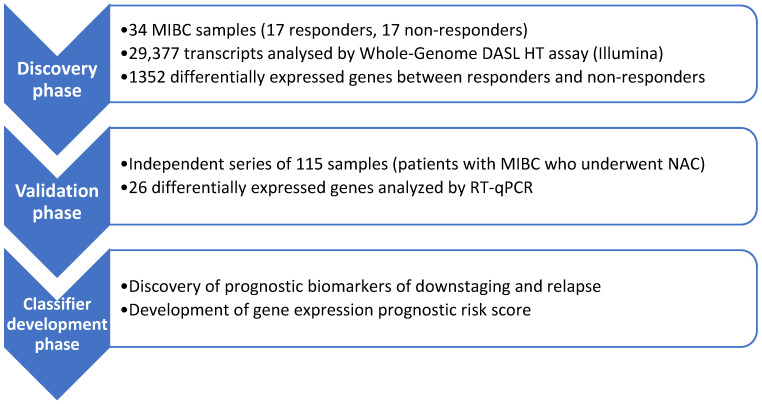
Flowchart of this study.

**Figure 2 biomedicines-13-00740-f002:**
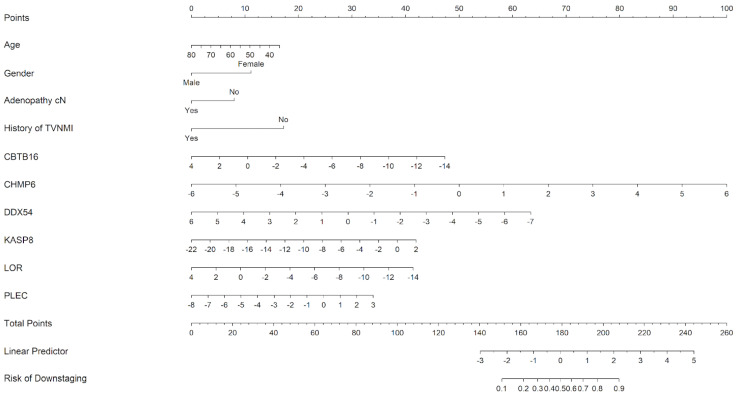
Nomogram for NAC response prediction based on 10 variables.

**Figure 3 biomedicines-13-00740-f003:**
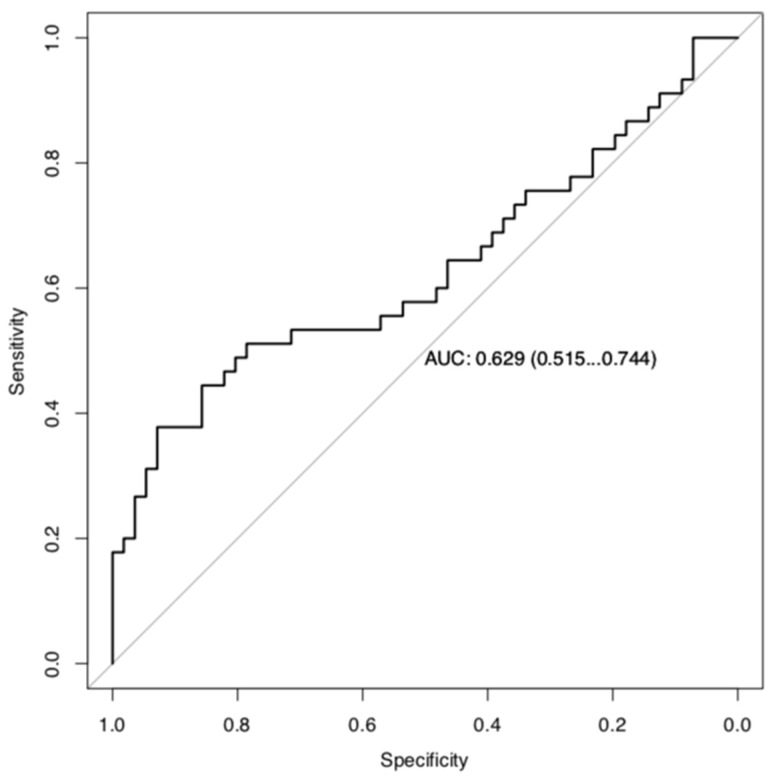
ROC curve for relapse prediction. Grey line represents an AUC of 0.5 indicating random guessing. Black line represents the performance of our proposed model.

**Figure 4 biomedicines-13-00740-f004:**
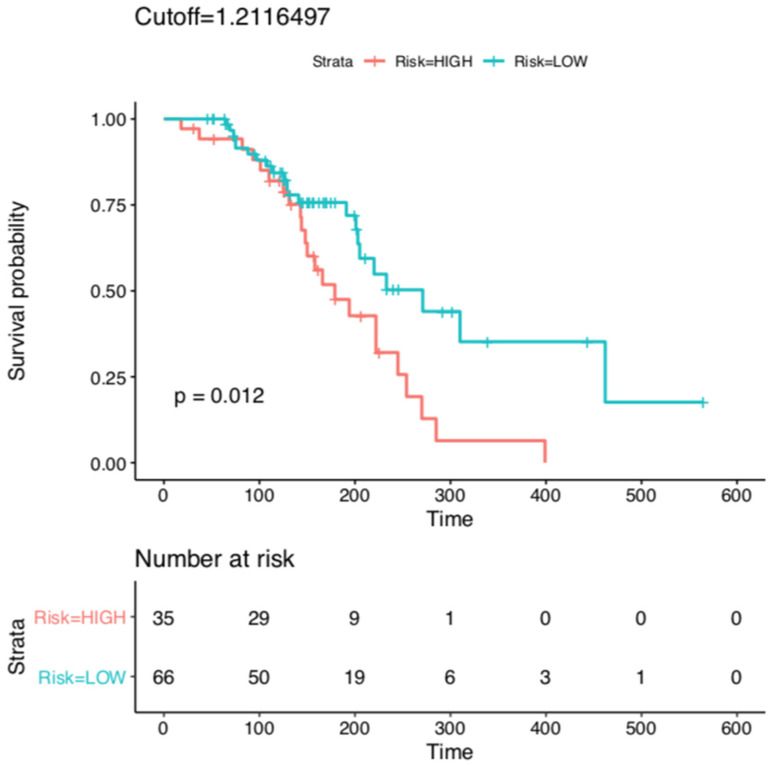
Kaplan–Meier plot for relapse-free survival based on the risk scores generated by our model.

**Table 1 biomedicines-13-00740-t001:** Demographic and clinicopathological features of the study population.

	Discovery Phase (n = 34)	Validation Phase (n = 115)
Age at diagnosis (years)	63.16	61.86
Gender (male/female)	31/3	104/11
NMIBC history (%)	20.5	27.8
Associated CIS (%)	17.6	7.8
Downstaging after neoadjuvant chemotherapy (%)	47	44.3
Pathologic stage		
T0	7 (20.5)	32 (27.8)
Cis	2 (0.05)	9 (7.8)
Ta	3(0.08)	2 (1.7)
T1	4(11.7)	8 (6.9)
T2	7(20.5)	18 (15.6)
T3	8(23.5)	24 (20.8)
T4	3(8.8)	22 (19.1)

Data are shown as n (%) or mean (SD).

**Table 2 biomedicines-13-00740-t002:** Cox’s regression model with the variables of the nomogram.

Characteristic	HR ^1^	95% CI ^1^	*p*-Value
Age	0.97	0.94, 1.00	0.084
Adenopathy			
No	-	-	
Yes	0.54	0.29, 1.00	0.051
Sex			
Female	-	-	
Male	0.42	0.14, 1.25	0.12
History of NMIBC			
No	-	-	
Yes	0.27	0.11, 0.63	0.003
CBTB16	0.82	0.68, 0.98	0.031
CHMP6	1.90	1.34, 2.71	<0.001
DDX54	0.69	0.53, 0.88	0.004
KASP8	1.14	1.05, 1.25	0.003
LOR	0.84	0.70, 1.00	0.048
PLEC	1.27	0.96, 1.68	0.10

^1^ HR = Hazard ratio, CI = Confidence Interval.

**Table 3 biomedicines-13-00740-t003:** Univariate analysis for predicting relapse.

	Univariate HR (95% CI)	*p*
FN1	0.85 (0.73–1)	0.05
VASP	0.85 (0.71–1.01)	0.06
CEP63	0.83 (0.70–0.99)	0.03

HR = hazard ratio; CI = confidence interval.

## Data Availability

The datasets used and/or analysed during the current study are available from the corresponding author upon reasonable request. The data are not publicly available due to privacy restrictions.

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
