# Peer review of "Biomarker-Based Nomogram to Predict Neoadjuvant Chemotherapy Response in Muscle-Invasive Bladder Cancer"

_biomedicines, 2025, doi:10.3390/biomedicines13030740_

Round 1

Reviewer 1 Report (Previous Reviewer 1)

Comments and Suggestions for Authors

The authors have significantly improved the manuscript. The revised version may be considered for publication.

Comments on the Quality of English Language

A bit of language refinement will enhance the interest of the readers. 

Author Response

The authors have significantly improved the manuscript. The revised version may be considered for publication.

We appreciate the acceptance of the paper by the reviewer. 

A bit of language refinement will enhance the interest of the readers. 

The introduction and discussion sections have been slightly refined.

Reviewer 2 Report (New Reviewer)

Comments and Suggestions for Authors

This manuscript entitled “Biomarker-Based Nomogram to predict Neoadjuvant Chemotherapy response in Muscle-Invasive Bladder Cancer” describes the establishment of a predictive nomogram model for bladder cancer. Below are my minor comments:

1. Introduction section:

1) Include some basic introduction to bladder cancer as well as the latest global statistical data.

2) Expand the discussion on neoadjuvant chemotherapy for bladder cancer, including current strategies and approaches.

2. Revise Figure 1 by including the criteria for sample selection and exclusion.

3. For Figures 2 and 4, it would be better if the authors could increase the font sizes of some plots, as some words are too tiny. In addition, improve the figure legends by detailing what each sub-panel represents.

4. For Figure 3, enhance the figure legend by explaining the meaning of the AUC values presented.

5. The abbreviation list does not fully cover all the abbreviations used throughout the manuscript. Authors should check and revise the list carefully.

Author Response

Thank you very much for your detailed and helpful advise. We have tried to use all your inputs to improve the paper and we think the result is optimal. Please find the reponses to every suggestion:

1) Include some basic introduction to bladder cancer as well as the latest global statistical data.

We have included more epidemiological information in the introduction section.

2) Expand the discussion on neoadjuvant chemotherapy for bladder cancer, including current strategies and approaches.

We have included a brief introduction on the current chemotherapy regimens according to the reviewer’s suggestion.

  1. Revise Figure 1 by including the criteria for sample selection and exclusion.

We appreciate the comment regarding figure 1. However, the selection and exclusion process both of samples and genes is complex enough to be summarized in the flow chart figure. All the detailes process is clearly defined in the text and we think putting it again in the figure would make the article difficult to understand.

  1. For Figures 2 and 4, it would be better if the authors could increase the font sizes of some plots, as some words are too tiny. In addition, improve the figure legends by detailing what each sub-panel represents.

We have increased the size of the whole Figures. We think it is now better. We also clarified on the Figure caption 4 the different sub-panel plots to make it clearer as suggested by the reviewer.

  1. For Figure 3, enhance the figure legend by explaining the meaning of the AUC values presented.

We have explained the meaning of the AUC within the text

  1. The abbreviation list does not fully cover all the abbreviations used throughout the manuscript. Authors should check and revise the list carefully.

The abbreviations list has been revised ans some missing abbreviations added.

Round 2

Reviewer 2 Report (New Reviewer)

Comments and Suggestions for Authors

The authors have addressed my comments accordingly.

This manuscript is a resubmission of an earlier submission. The following is a list of the peer review reports and author responses from that submission.

Round 1

Reviewer 1 Report

Comments and Suggestions for Authors

Title:  Biomarker-Based Nomogram to predict Neoadjuvant 2 Chemotherapy response in Muscle-Invasive Bladder Cancer

Manuscript ID: biomedicines-3484122

Journal: Biomedicines

  1. Abstract section Line No 42, clarify what the author means by T is and check the spelling of lymphadenopathy; (age, T is, gender, history of NMIBC, limfadenophaty----).
  2. In the introduction section, line 53, the authors write, “In fact, half of these patients will die within 2 years from diagnosis despite undergoing radical cystectomy (RC).” The sentence can be better expressed as “half of these patients may die within 2 years from diagnosis despite undergoing radical cystectomy (RC).”
  3. In the next sentence, they write, “Neoadjuvant cisplatin-containing combination chemotherapy is considered an strong recommendation for treating T2-T4N0Mo MIBC.” The usage of the article is incorrect and they should give a brief description of T2-T4N0Mo MIBC”
  4. Introduction section, line Nos 59-63, “The most recent meta-analysis encompassed four additional randomized controlled trials (RCTs), incorporating fresh data from 427 patients along with updated results from 1,596 participants in the Nordic I, Nordic II, and BA06 30894 trials. This analysis reinforced the findings of earlier studies, indicating an 8% absolute increase in five-year survival rates, with a need to-treat ratio of 12.5, and a 30-40% of complete pathological response [2].” The given citation is from 2005, not that recent.
  5. Lines Nos 64-66, they are writing the sentence in future tense. However, I perceive that this is the finding of some other research work that they try to quote. “However, those muscle-invasive bladder cancer (MIBC) patients not responding to NAC will experience what could be considered a delay on the surgical treatment and face the potential chemotherapy side effects.”
  6. The introduction section needs to be rechecked and written properly.
  7. In Tables 1-3, Check the comma usage in the data instead of a decimal point. Also, check the data mentioned in the text like for example, in lines Nos, 224-226, “The median range value of this risk score value was 0,85 (0,73-1). Thereafter, a ROC analysis of this combined gene expression model allowed the selection of a threshold of 1,211 to classify patients into a high and a low risk group for relapse.”
  8. Table No1. The parameter “Age at diagnosis, years” should be written as “Age at diagnosis (years).”
  9. How can the findings of this study be translated to clinical relevance?
  10. The manuscript should be thoroughly checked for grammatical and typo errors.
  11. The manuscript needs to be checked thoroughly.
Comments on the Quality of English Language

There are grammatical errors, specifically related to the use of articles and tenses. Also, the manuscript needs to be checked for typo errors.

This causes difficulty in understanding what the authors try to convey.

Reviewer 2 Report

Comments and Suggestions for Authors

General comment

The manuscript entitled “Biomarker-Based Nomogram to Predict Neoadjuvant Chemotherapy Response in Muscle-Invasive Bladder Cancer” presents a retrospective multicenter study that aims to develop a nomogram integrating gene expression and clinical variables to predict response to neoadjuvant chemotherapy (NAC) in muscle-invasive bladder cancer (MIBC). The authors analyze gene expression patterns using microarrays and validate selected genes via RT-qPCR. A prognostic model is constructed based on gene expression data and clinical factors. Despite the interesting topic there are few major concerns that limits the work:

-          The study suffers from significant methodological flaws. The retrospective nature introduces selection bias, and the inclusion criteria are not well-defined, particularly regarding the exclusion of cases due to unavailable tissue samples. The authors fail to provide a power calculation to justify their sample size.

-          The study lacks a robust validation cohort, which limits the generalizability of the findings.

-          The ROC analysis for relapse prediction yields an AUC of 0.629, which is barely above random chance (0.5). This suggests that the model lacks clinical utility.

-          The genes selected for the final model (CBT16, CHMP6, DDX54, CASP8, LOR, PLEC) have not been previously established as predictive biomarkers in bladder cancer, and their biological relevance to NAC response is not convincingly demonstrated.

-          The proposed nomogram does not provide a meaningful improvement over existing clinical models that incorporate standard prognostic factors such as tumor stage, lymph node involvement, and histopathological features.

Unfortunately, given the significant methodological flaws,  weak predictive performance as well as lack of external validation, the study requires substantial revisions.